# Mechanical Properties and Biocompatibility of 3D Printing Acrylic Material with Bioactive Components

**DOI:** 10.3390/jfb14010013

**Published:** 2022-12-23

**Authors:** Zbigniew Raszewski, Katarzyna Chojnacka, Julita Kulbacka, Marcin Mikulewicz

**Affiliations:** 1SpofaDental, Markova 238, 506-01 Jicin, Czech Republic; 2Department of Advanced Material Technologies, Faculty of Chemistry, Wroclaw University of Science and Technology, Smoluchowskiego 25, 50-372 Wroclaw, Poland; 3Department of Molecular and Cellular Biology, Faculty of Pharmacy, Wroclaw Medical University, Borowska 211A, 50-556 Wroclaw, Poland; 4Department of Dentofacial Orthopaedics and Orthodontics, Division of Facial Abnormalities, Wroclaw Medical University, Krakowska 26, 50-425 Wroclaw, Poland

**Keywords:** 3D printing, bioactive glass, ion release, mechanical properties

## Abstract

The aim of this study was to create a 3D printing material with bioactive properties that potentially could be used for a transparent removable orthodontic appliance. Materials and methods. To acrylic monomers, four bioactive glasses at 10% concentration were added, which release Ca, P, Si and F ions. The materials were printed on a 3D printer and tested for flexural strength (24 h and 30 days), sorption and solubility (7 days), ion release to artificial saliva pH = 4 and 7 (42 days) and cytotoxicity in the human fibroblast model. The released ions were determined by plasma spectrometry (Ca, P and Si ions) and ion-selective electrode (F measurement)s. Results: The material obtained released Ca^2+^ and PO_4_^3−^ ions for a period of 42 days when using glass Biomin C at pH 4. The flexural strength depended on the direction in which the sample was printed relative to the 3D printer platform. Vertically printed samples had a resistance greater than 20%. The 10% Biomin C samples post-cured for 30 min with light had a survival rate of the cells after 72 h of 85%.Conclusions. Material for 3D printing with bioactive glass in its composition, which releases ions, can be used in the production of orthodontic aligners.

## 1. Introduction

In recent years, the market has seen an increasing share of materials for the production of prosthetic restorations that are printed by 3D printing. They can be used to make, among others, temporary crowns and bridges, surgical templates, removable orthodontic appliances, and dentures [1].

3D printing undoubtedly has many advantages over the traditional process of forming orthodontic appliances. First, there is no need to make models (additional material). In the thermoforming process, waste is additionally generated from the foil that has been pressing on the model. Furthermore, this new technology can shorten the time to make a single appliance and reduce laboratory labor costs [2].

However, a number of publications indicate that 3D printed materials have lower mechanical properties than products made with traditional heat-curing resins or CAD CAM technology [3]. To solve this problem, investigations are underway in two main areas. The first direction is the development of materials for 3D printing with increased mechanical properties by adding nanofillers, e.g., ZrO_2_, [2,4] cellulose nanocrystals, and silver nanoparticles [5,6].

The second activity focuses on the development and synthesis of new acrylic oligomers that possess better mechanical properties and less shrinkage during polymerization [7].

In the literature, it is possible to find extensive information on the mechanical properties tested in the course of research on materials suitable for 3D printing and their biocompatibility. However, these are mostly commercial products, allowed for sale, but their exact composition remains the manufacturer’s secret. Therefore, an important piece of information for the reader may be to perform a research model with methacrylate resins and test their properties [8].

Another important consideration for 3D materials is their contact time with the patient’s oral cavity. Removable clear aligners are in direct contact with the patient’s mouth for a long period of time (22 h/day). If the patient does not clean his teeth after eating, ideal conditions are created for the demineralization of the enamel through the action of sugar-decomposing bacteria. For this group of users, it would be good if this material had bioactive properties. On the one hand, these properties prevent the formation of biofilms on their surface, but also provide ions that could be used to initiate the remineralization process of the tooth that is in direct contact with the material [9]. Additional release of cations can locally raise the pH above 5.5, which can significantly prevent the early stages of caries [10].

In materials that have a longer contact with the patient, a very important parameter is their biological compatibility (it must not be cytotoxic, irritating, or allergic) [11]. However, polymerization during 3D printing causes only a partial reaction of the double bonds and a certain degree of conversion of the C=C double bonds in the methacrylic group. For this reason, for the material to be fully cured and more bio-compatible, it is necessary to place the material in a light furnace for the post-curing process.

The curing time of the material after 3D printing is of great importance for its properties. It depends, among other things, on the color of the material, the single thickness of the printed layer, and the composition of the resin itself. Materials with acrylates in their composition undergo faster cross-linking reactions and show a lower value of oxygen inhibition [12]. Cycloaliphatic and aromatic acrylates show less shrinkage compared to the standard monomer with low molecular mass, such as triethyl glycolmethacrylate or 2- hydroxy ethyl methacrylate. Shrinkage and associated stress can influence the deformation during 3D printing when the object is created layer by layer [13].

All these parameters must be taken into account when creating a new 3D printing material. During this research, it was hypothesized that there is a possibility of creating a material with the possibility of long-term ion release that can be used in a 3D printer. In our previous work, we managed to create a material with similar properties to those of the materials that used poly (methyl methacrylate) as an organic matrix. As 3D printing materials contain higher methacrylates, it is necessary to confirm this hypothesis for other compounds of this group as well [14,15].

## 2. Materials and Methods

### 2.1. Material Preparation

The test glass samples were provided by Cera Dynamic (Kent, England). They were made by mixing certain oxides with each other, heating to a temperature of 1500 °C and then rapidly cooling them in water. Subsequently, these powders were ground to a granulation of 5 microns. The samples obtained in this way were used for further tests without modification. The following glasses were used for the test: Biomin C, Biomin F, S53P4, 45S5 [14]. The detailed composition of each glass was described in our previous publication.

The most common raw materials (meth) of acrylate used in 3D printing are polyethylene glycol dimethacrylate, urethan dimethacrylate, triethylene glycol dimethacrylate (TEGDMA), bisphenol A-glycidyl methacrylate (Bis-GMA), trimethylolpropane triacrylate (TTA) and bisphenol A ethoxylate diacrylate (Bis-EDA) [13]. Therefore, it was decided to use this type of material to create a model material for 3D printing.

Acrylate oligomers were mixed in a dark glass bottle with a magnetic laboratory stirrer, and 500 g of material was prepared for all experiments. The composition is presented in Table 1.

Subsequently, this material was divided into five other dark PE packages and 10 g of bioactive glass and 5 mm diameter ceramic balls (100 g) were added to 4 of them. Each material was mixed for 60 min in a small laboratory mixer (Izerska Porcelanka, Izera, Czech Republic) to obtain a homogeneous suspension of glass particles in the whole material. Ceramic balls were removed and material was used immediately for the 3D printing process.

Samples for further tests were printed on the Liquid Crystal Precision (Photocentric, Peterborough, UK) 3D printer. To test the flexural strength, 120 test pieces with dimensions 64×10×3.3 mm were prepared. For the sorption, solubility and ion release tests, the samples had a diameter of 15 and a thickness of 1 mm (90 samples totally). To determine cytotoxicity in human fibroblasts, 15 samples of dimensions 5 × 5 × 1 mm were made.

After removing the samples from the printer, they were placed in an isopropyl alcohol solution and washed ultrasonically for 10 min. Then they were cured in a light chamber (Evicrobox, SpofaDental, Jicin, Czech Republic) for 30 min. The support structures were removed with a prosthetic micromotor, and the uneven edges were polished with 30-micrometer grit paper. The scheme of execution of all samples of the tests carried out is presented in Figure 1.

### 2.2. Extraction of Unpolarized Monomers and Catalyst

The idea of testing unpolymerized methacrylic monomers and resin catalysts was proposed by González et al. [16]. They used ethanol as an extraction medium and UV-vis spectroscopy for detection to check the hardening of the resin with light.

Samples with a diameter of 15 mm and thickness of 1 mm (30 samples total) after removal from the 3D printer were polymerized in an Evicrobox lamp for 10 or 30 min. Then each sample was placed in a dark glass jar and 10 mL of ethanol was added. The contact time was 2 h. After this period of time, the alcohol solution was analyzed with a UV-vis spectrophotometer in the wavelength range of 190–400 nm (Helios, Thermo Fisher Scientific, Waltham, MA, USA). In this wave range, absorption of catalysts and acrylate monomers with aromatic rings in their structure is visible.

### 2.3. Flexural Strength

Articles describing 3D materials forward the thesis that the mechanical properties of printed material depend on the orientation of the printed object [3]. The second important parameter that influences the mechanical properties is the time of post-cured samples in the light-curing chamber. Therefore, the strength tests during this investigation were performed with material polymerized for 30 min.

To confirm the thesis regarding the location of the building object during printing for the flexural strength test, the samples were printed in two directions: horizontally and vertically to the printer platform (Figure 2).

In total, 120 samples were printed, with dimensions of 64 × 10 × 3.3 mm [14]. Mechanical resistance to fracture was tested using three-point deflection (50 mm supports) with a breaking head speed of 5 mm/min. A detailed description of this test can be found in ISO 20795-1: 2013 (en). Dentistry—Denture base polymers [14].

Samples containing pure polymerized resin and bioactive glass were stored with distilled water at 37 °C in a laboratory dryer. After 24 h, the first part of sample 6 of each type was placed in an Autograph tensile strength instrument (Shimadzu, Duisburg, Germany) and a three-point fracture resistance test was performed. The other part of the samples was subjected to the same test after a period of 30 days. The distilled water in the samples stored for 30 days was changed every 3 days.

### 2.4. Sorption and Solubility

As the orthodontic aligner is located in the oral cavity, where it is exposed to various types of water-based solutions, sorption and solubility are important parameters determining the whole development.

This research was to determine the quality of the residuals that can be washed out of the storage of the material in distilled water for 7 days and the amount of water absorbed by the material. This test was carried out according to ISO 20795-1: 2013 (en). Norm [17].

In the beginning, the samples were placed in a desiccator (Schott, Wolverhampton, England) until the mass became constant (*M*1), then immersed in distilled water at 37 °C. After 7 days, the tested materials were removed from the solution and weighed again, determining the mass of the wet sample (*M*2). For the rest of the test, the testing discs were dried in a desiccator until the mass was steady for 14 days (*M*3). For the calculation of sorption and solubility, the following equations were used:(1)$$\mathrm{Sorption}\;[µg/{\mathrm{mm}}^{3}]\;=\;\frac{M2-M1}{V}$$
(2)$$\mathrm{Solubility}\;[µg/{\mathrm{mm}}^{3}]\;=\;\frac{M1-M3}{V}$$
where *V* is the volume [17].

### 2.5. Ion Release

To test the bioactive properties of the samples, which can be verified as the ions released to artificial saliva, we tested two pH values (4 and 7) during 1, 28 and 42 days. Discs with a diameter of 5 mm and a thickness of 1 mm were prepared for the tests. The sample preparation method is described above. The total number of samples in this study was 15 (3 for each type of material). Disks made of 3D printing resin were used as reference material.

The saliva extracts of the samples were analyzed using the inductively coupled plasma atomic emission spectrometry method using the iCAP 6500 Duo optical spectrometer with horizontal and vertical plasma (Thermo Fisher Scientific, Waltham, MA, USA). A detailed description of the preparation of the artificial saliva solution and the extraction method over time at different pH levels is described in our previous article [14].

### 2.6. Cell Culture

After the initial test of the mechanical properties of the material after the addition of bioactive glass, the samples containing 10% Biomin C and a resin (as a reference) were selected considering the highest mechanical resistance of this material for the cytotoxicity study. The effect of these two materials on cell cultures was then investigated.

The test samples before the experiments were polymerized for 30 min in a light chamber and washed with isopropanol ultrasonically for 10 min.

In the cytotoxicity study, human gingival fibroblasts (HGF) were used. This primary cell culture was isolated from a fragment of gingival tissues previously described by Saczko et al. [18]. Cells were grown in DMEM (Dulbecco’s Modified Eagle’s Medium, Sigma-Aldrich, Poznan, Poland) supplemented with 10% FBS (Sigma), penicillin/streptomycin (Sigma) and 1% of GlutaMAXTM-I (Gibco, New York, NY, USA) in 25 cm^2^ flasks (Zeit Buddels, Sarstedt, Germany). The cells were maintained in a humidified atmosphere at 37 °C and 5% CO_2_. For experimental procedures, cells were removed by trypsinization (0.25% Trypsin/EDTA, IIDT, Wroclaw, Poland).

The Evaluation of Cell Morphology and Migration as a Marker of Cytotoxicity 

This method visualizes the direct impact of verified material on target cells. Cells were grown in standard Dulbecco Modified Medium (DMEM, Sigma) in the presence of the tested articles. For microscopic observations, the tested articles were placed on a 24-well plate (Nunc), then HGF cells were seeded at a density of 1 × 104 cells/well in a culture medium. This set-up was maintained in a humidified atmosphere at 37 C and 5% CO_2_. Microscopic observations were made after 1 h and 24 h of exposure to the tested articles (Leica DMi 1, CellService, Poznan, Poland) [19].

### 2.7. Cytotoxicity Assay Test—Direct Contact

For the cytotoxicity studies, 4 samples with a diameter of a minimum 5 mm and a thickness of 1–2 mm were prepared. The impact of the test articles on gingival fibroblasts was evaluated after 24, 48 and 72 h exposure using the PrestoBlue^®^ assay (ThermoFisher, Warsaw, Poland). Before exposition, the tested articles were placed on a 24-well plate (Sarstedt, Germany), then HGF cells were seeded at a density of 1 × 10^4^ cells/well in a culture medium. After 24, 48, or 72 h, the culture medium was taken for PrestoBlue measurements. PrestoBlue^®^ is a ready-to-use cell-permeable resazurin-based solution that functions as a cell viability indicator by using the reducing power of living cells to quantitatively measure the proliferation of cells. The absorbance was measured on a multiplate reader (GloMax^®^ Discover, Promega, Madison, WI, USA) using a 560 nm wavelength for signal detection. The exposure was performed in triplicate for each test article.

The results obtained from the measurements defined as viability were calculated in comparison to the values of control untreated cells. In the calculations, mean values were used [19].

Statistical analysis was performed with a one-way ANOVA using the Tukey HSD test calculator available at Astasta.com. For all tests, the confidence level was assumed at *p* < 0.05

## 3. Results

### 3.1. Extraction of Unpolymerized Monomers and Catalyst

The polymerized material has a residual monomer in its composition after the light-curing process. Extracts of non-polymerized materials are shown in Table 2.

The results obtained indicate that extending the exposure time in the light-curing chamber from 10 to 30 min reduces the content of the eluted components by 50–60%. The compound diphenyl (2,4,6-trimethylbenzoyl) phosphine oxide (light-curing catalyst) contributes to absorption at a wavelength of 372 nm. The maximum absorption for wavelength 271 is the residual stabilizer 2,6-Di-t-butyl-p-cresol, which is used in methacrylate resins to prevent self-polymerization. The printing material includes ethoxylated bisphenol A dimethacrylate, whose maximum absorption is 265 nm. After a longer period of exposure to the sample in a light furnace, the greatest reduction in the amount of the ingredient is observed, because part of the monomer is polymerized into the material structure.

### 3.2. Flexural Strength

The results of flexural strength after 24 h and 30 days of storage in distilled water are presented in Table 3.

It is possible to notice differences in flexural strength between the samples that were printed horizontally and vertically to the printer platform. The first group of samples had a 20% lower resistance to fracture. When stored in distilled water for 30 days, the materials weakened due to the effect of the water plasticizer. The highest resistance to breaking was obtained for the pure methacrylic resin samples, and the second highest was for the sample containing 10% Biomin C glass.

### 3.3. Sorption and Solublity

The effects of water on the properties of the samples are presented as sorption and solubility in Table 4.

When placed in water, the samples dissolved partially and absorb the water. This is visible in the samples containing Biomin C (sorption 140.48 ± 2.76 μg/mm^3^). For other samples, these sorption values are eight times higher than for samples containing only resin. The solubility of a sample with bioactive glass alone ranges from 1–3 μg/mm^3^.

The visual chaining in the samples shown in Figure 3 indicates that the material under the influence of storage in water is hydrolyzed, which can be observed as a change in color and lack of transparency. In Figure 4, the material sample containing Biomin F glass, surface changes are visible at magnification. The roughness of the sample surface increased.

With knowledge of the mechanical properties of the material, its biocompatibility was tested in the second part of the study.

### 3.4. Cytotoxic Study

The biocompatibility study showed that both tested materials were not cytotoxic to cells, and that the morphology of cells was not affected. The reference resin material slightly changed the morphology of the fibroblasts, causing shrinkage, but this did not affect cell viability. The +10% Biomin C samples were completely safe for cells (Figure 5 and Figure 6).

Cell viability after 24, 48 and 72 h of application of the test samples of resin and resin +10% Biomin C was >70% (Figure 5).

The tests carried out in cell cultures indicate that the material prepared for 30 min in a light furnace and washed with an alcohol solution does not have cytotoxic properties in terms of cell cultures.

### 3.5. Ion Releasing

In the last stage of the research, it was decided to test the ions released from a sample of bioactive glass in a polymerized material. The results obtained with coupled plasma atomic emission spectrometry are presented in Table 5.

Biomin C glass samples subjected to the elution test may release calcium ions over a long period of time. In an acid environment, this reaction was faster at pH = 4. Biomin F glass samples released fluoride ions, but the process was very fast. All glass specimens had the ability to release phosphate anions over time.

## 4. Discussion

The thesis put forward at the beginning of this study has been confirmed, and it is possible to create a material that can deliver calcium cations, as well as phosphate and silicate anions, and, in a short period of time, fluorine. A certain restriction for this material may be its flexural strength. Samples stored in distilled water for 30 days had a lower fracture resistance than ISO < 60 MPa. The flexural strength of the 3D printed resin is one of the most tested parameters, as it is considered a major type of clinical failure.

The results obtained in this study indicate two important issues. Fracture resistance depends on the orientation of the sample during printing along the plane of the printer table. Horizontally arranged samples had a mechanical resistance of 20–25% lower than those of printed verticals.

An explanation of this phenomenon is provided in the article by Gad et al. During printing, the resin is in contact with the air dissolved in it, particles of which can be obtained between the individual printed layers. This creates voids that can significantly reduce the bending resistance of the material [3,8,19,20].

The mechanical properties of the material are additionally influenced by the time it takes to post-cure the sample in the light furnace. When the curing time is too short, the material has a high content of residual monomers, which act in a similar way as plasticizers, reducing the material’s resistance to flexing. When printing vertically to the printer platform, you should pay attention to one more issue: the smoothness of the printout obtained, which may be lower in this sample arrangement. As presented by Alshaikh et al., to eliminate this, it seems necessary to polish the restoration better [2].

The second important issue is the degree of conversion of the double bonds in the methacrylic bond; similarly, as in the case of composites, it may be on the level of 50–80% in the material post-cured in a light furnace. This can significantly affect the mechanical resistance of the material [21].

Current research shows the effect of exposure time in a furnace on the content of residual monomers. Samples polymerized for 30 min have a twice lower content of ethoxylated dimethacrylate, which was extracted with ethanol and measured with UV-vis spectrophotometry.

The addition of bioactive glass also reduced the mechanical properties. This is due to the fact that the glasses used were not subjected to the silanization process and therefore they did not produce a chemical connection between the resin and the filler.

Removable orthodontic aligners are in contact with the oral cavity for 7–14 days, depending on the manufacturer. Therefore, the sorption and solubility of this type of material are important. The results of these tests indicate that the material containing bioactive glass undergoes a very high water absorption. This is necessary to initiate the hydrolysis of the glass contained in the sample and the evolution of ions. The mean sorption of the resin sample was 2% of the sample mass, which is comparable to the results obtained by Altarazi et al. [8].

Unpolymerized acrylic resins are irritating and highly cytotoxic. Furthermore, hydrolysis of the residual monomer in water can produce methacrylic acid, which can decrease the local pH and adversely affect physiological properties [13].

In this study, the theory of the influence of the length of direct exposure time on the biological properties of the material was also confirmed. Thirty-minute exposure to the light furnace causes a reduction in the concentration of residual monomers, and the resin and sample with 10% Biomin C did not have a cytotoxic effect on cell cultures of human gingival fibroblasts. Aati et al. reached similar conclusions in their research when the post-curing time was 20 min [22].

The 3D printing materials prepared in this study can release ions over a longer period of time. They are mainly calcium cations and phosphate anions. Fluorine ions from a sample containing 10% Biomin F were very quickly washed out by artificial saliva at pH 4 and 7.

In our previous article, we described the release of ions from the same four types of glass in samples containing polymethyl methacrylate resins [14].

In our previous study, calcium ion release occurred in a smaller amount, e.g., in the case of Biomin C, it was 2.42 mg/L after 42 days (pH = 4.0) and 1.19 mg/L (pH = 7). However, when the same glass was added to the dimethacrylate-containing material, the release of cations was much more intensive after the same period of time (65.75 (pH = 4.0) and 20.14 mg/L (pH = 7.0)).

Fluoride ions, independently of the organic matrix (3D printing material and PMMA—previous publication), were released from the sample in the first 24 h. However, in the case of 3D printing material, this occurs at the same level in an acid and neutral environment.

A very important fact is that all samples can deliver phosphate anions in the amount of 20–44 mg/L over a period of 42 days.

pH has a very large influence on ion release from the glass. The process also takes place faster in an acidic environment than in a neutral pH. This is very important from a clinical point of view because cations can neutralize the acids created by bacteria and stop the process of demineralization of the enamel.

The obtained results of ion migration are similar to those presented by Al-Eesa and Liu et al., who studied ion release from the same glasses but in the case of composite materials. They proved by roentgenography that a layer of hydroxyapatite is formed on the surface of the material after long contact with water [23,24,25,26].

The rate of ion release, in addition to the composition of the polymer matrix, is influenced by the large size of the bioactive particles. They cannot be too small because the hydrolysis then takes place too quickly. If they are too large, the rate of ion secretion decreases [27,28].

More research should be conducted towards better prosthetic solutions, both in 3D technology and traditional prosthetic solutions [29,30,31,32].

## 5. Conclusions

The thesis presented at the beginning of our research has been confirmed. It is possible to make a 3D printing material prepared with 10% addition of various active glasses that can release calcium, silicon and phosphate ions.In the case of Biomin F glass, the release of fluorine ions in an acidic and neutral environment was very dynamic (it occurred within the first 24 h).Acrylic resins modified with 10% Biomin C glass can be valuable sources of calcium cations and phosphate anions under acid and neutral conditions over a period of 42 days.More research is needed on the composition of the methacrylate resins used to create the material for 3D printing, which would have a higher mechanical resistance to fracture.

## Figures and Tables

**Figure 1 jfb-14-00013-f001:**
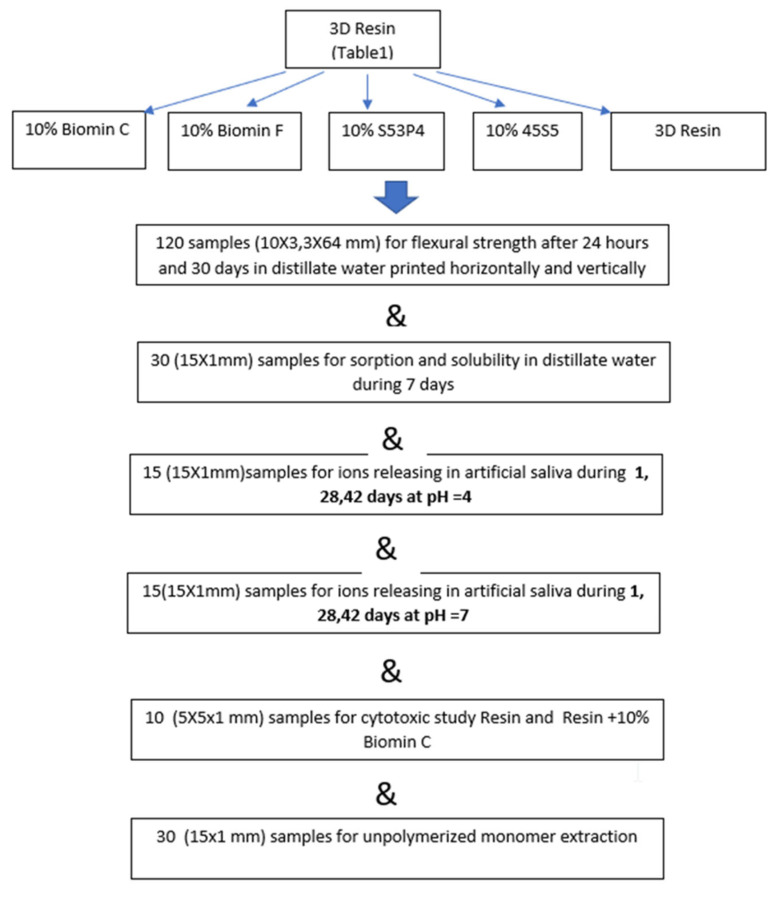
Scheme of sample preparation and testing in this work.

**Figure 2 jfb-14-00013-f002:**
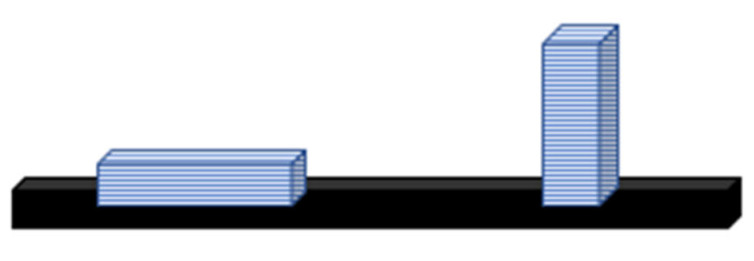
Directions of printing samples on the platform. (**left**) horizontal, (**right**) at an angle of 90°.

**Figure 3 jfb-14-00013-f003:**
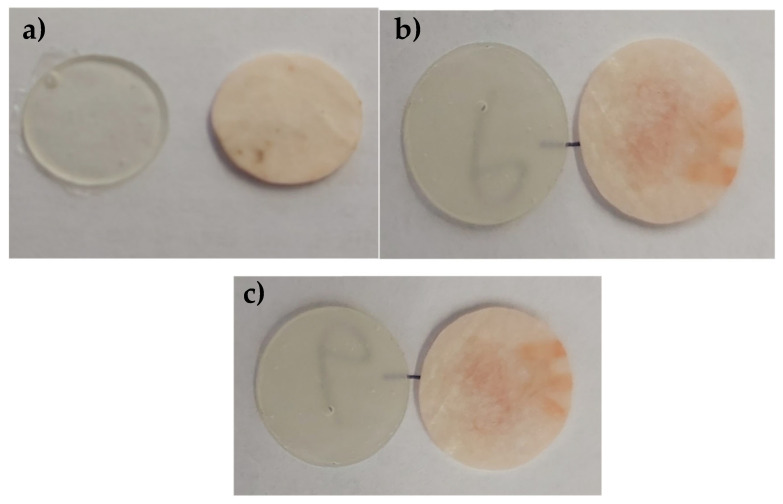
Samples of material (diameter 15 mm), before immersion in water (left) and after 7 days in distillate water (**a**) S53P4, (**b**) 45S5, (**c**) Biomin F.

**Figure 4 jfb-14-00013-f004:**
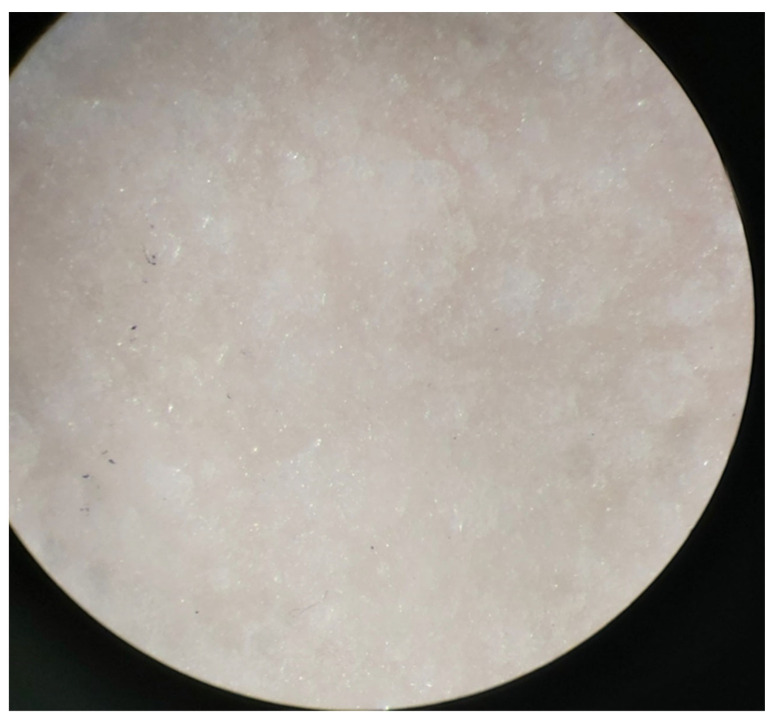
Biomin C surface, diameter 15 mm, after immersion in water, magnification 25×.

**Figure 5 jfb-14-00013-f005:**
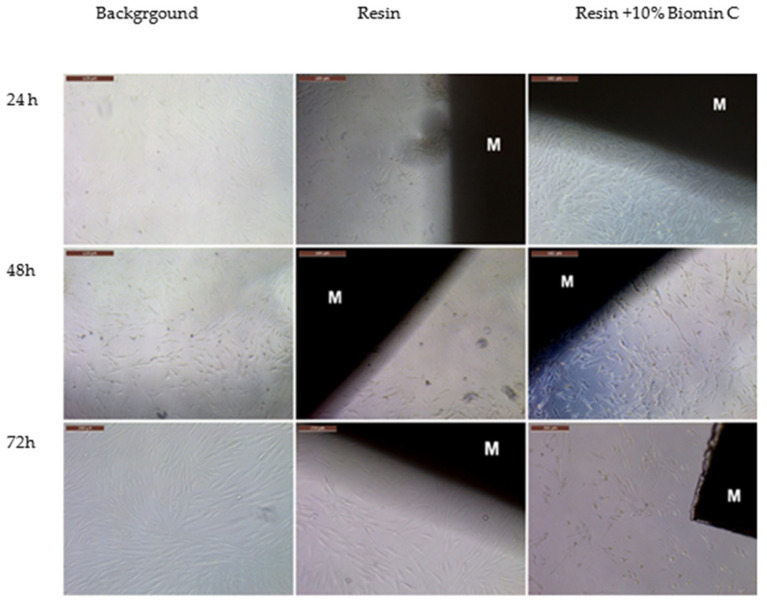
The biocompatibility assay on human gingival fibroblasts (HGFs) from the primary culture after 24, 48 and 72 h of incubation (magnification 5 or 10×, Leica). M—tested material. (The brown stripe in the upper left corner = 500 µm).

**Figure 6 jfb-14-00013-f006:**
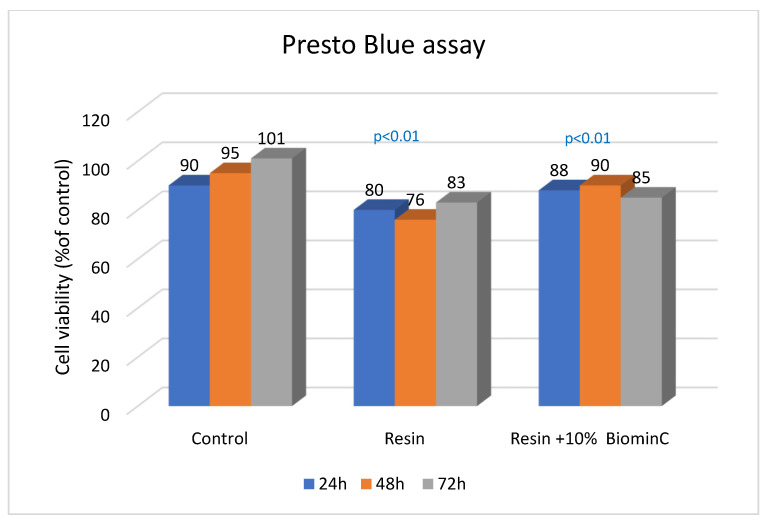
Cell viability over time for the tested samples. *p*—calculated for control samples.

**Table 1 jfb-14-00013-t001:** Composition of 3D printing material.

Monomer	Concentration [%]	Producer
Ethoxylated (E4) bisphenol A dimethacrylate (E4GMA) 98%	60%	Esschem
7,9,9-trimethyl-4,13-dioxo-3,14-dioxa-5,12-diazahexadecane-1,16-diyl bismethacrylate (UDMA) 98%	25%	Evonik
Triethylene glycol dimethacrylate (TEGDMA) 99%	10%	Evonik
Pentaerythritol tetraacrylate 96%	3%	Sigma-Aldrich
diphenyl(2,4,6-trimethylbenzoyl) phosphine oxide 98%	1%	Sigma-Aldrich
silicon dioxide Aerosil R972 99%	1%	Evonik

**Table 2 jfb-14-00013-t002:** Extract of the materials polymerized in a light chamber for 10 and 30 min after contact with ethanol for 2 h measured on the UV Vis Spectrophotometer (Absorbance).

		Resin	Resin + 10% Biomin C	Resin + 10% Biomin F	Resin + 10% S53P4	Resin + 10% 45S5
Time [min]	Wave Length [nm]	Absorbance ± SD	Absorbance ± SD	Absorbance ± SD	Absorbance ± SD	Absorbance ± SD
10						
	271	0.046 ± 0.011	0.056 ± 0.015	0.061 ± 0.013	0.054 ± 0.019	0.059 ± 0.009
	265	0.104 ± 0.011	0.131 ± 0.012	0.151 ± 0.024	0.144 ± 0.02	0.164 ± 0.022
	372	0.016 ± 0.011	0.018 ± 0.011	0.021 ± 0.011	0.026 ± 0.011	0.025 ± 0.011
30						
	271	0.016 ± 0.009 *p* < 0.01	0.023 ± 0.012 *p* < 0.01	0.023 ± 0.019 *p* < 0.01	0.026 ± 0.013 *p* < 0.05	0.031 ± 0.011 *p* < 0.05
	265	0.049 ± 0.015 *p* < 0.01	0.063 ± 0.015 *p* < 0.01	0.061 ± 0.02 *p* < 0.01	0.063 ± 0.023 *p* < 0.01	0.073 ± 0.026 *p* < 0.01
	372	0.07 ± 0.005	0.09 ± 0.004	0.012 ± 0.01	0.06 ± 0.09	0.008 ± 0.05 *p* < 0.01

**Table 3 jfb-14-00013-t003:** Flexural strength [MPa] after 7 and 30 days at 37 °C in distilled water.

		Resin + 10% S53P4	Resin + 10% 45S5	Resin + 10% Biomin F	Resin + 10% Biomin C	Resin
horizontally						
	24 h	66.42 ± 3.01	66.66 ± 3.02 *p* < 0.01	58.92 ± 4.68 *p* < 0.01	69.28 ± 2.94	73.2 ± 3.74
	30 days	54.1 ± 3.01	53.2 ± 4.03	52.72 ± 1.0	53.88 ± 1.73	67.81 ± 4.65
		*p* < 0.01	*p* < 0.01	*p* < 0.01	*p* < 0.01	
vertically						
	24 h	58.1 ± 4.41	47.34 ± 5.02	42.11 ± 3.33	43.32 ± 4.03	53.2 ± 4.03
	30 days	47.3 ± 5.98	*p* < 0.01 43.1 ± 3.66 *p* < 0.05	*p* < 0.01 38.13 ± 4.71 *p* < 0.01	*p* < 0.01 40.92 ± 6.01 *p* < 0.01	49.2 ± 5.43

*p**—calculated for pure resin samples.

**Table 4 jfb-14-00013-t004:** Sorption and solubility of resins with bioactive glass after 7 days in distilled water.

	Resin + 10% S53P4	Resin + 10% 45S5	Resin + 10% Biomin F	Resin + 10% Biomin C	Resin
solubility [μg/mm^3^]	3.09 ± 0.32 *p* < 0.01	1.46 ± 0.39 *p* < 0.01	1.58 ± 0.14 *p* < 0.01	1.18 ± 0.09	0.36 ± 0.09
sorption [μg/mm^3^]	71.39 ± 13.61 *p* < 0.01	89.38 ± 11.15 *p* < 0.01	83.93 ± 4.08 *p* < 0.01	140.48 ± 2.76 *p* < 0.01	11.06 ± 0.88

*p**—calculated for pure resin samples.

**Table 5 jfb-14-00013-t005:** Ion content extracted from glass samples at pH = 4 and pH = 7 after 1, 28 and 42 days.

	Ca	P	Si	F^-^
Sample Days	[mg/L]	[mg/L]	[mg/L]	[mg/L]
* ** Background saliva ** *				
**pH 4 0**	<LOD	19.14 ± 3.32	0.007300 ± 0.0	<LOD
**pH 7 0**	<LOD	20.65 ± 2.54	0.06050 ± 0.01	<LOD
* ** 10% S53P4 ** *				
**S1 pH4 1**	1.058 ± 0.012	40.22 ± 3.21	0.7269 ± 0.03	<LOD
**S1 pH4 28**	1.960 ± 0.19	30.41 ± 2.11	6.216 ± 0.36	<LOD
**S1 pH4 42**	0.4599 ± 0.011	39.78 ± 2.76	5.574 ± 0.39	<LOD
**S1 pH7 1**	0.593 ± 0.015	41.91 ± 1.95	0.7802 ± 0.061	0.1595 ± 0.009
**S1 pH7 28**	0.2698 ± 0.001	40.75 ± 2.74	3.135 ± 0.15	<0.10
**S1 pH7 42**	0.2445 ± 0.016	39.20 ± 1.32	3.836 ± 0.016	0.1347 ± 0.007
* ** Resin ** *				
**S2 pH4 1**	<LOD	19.24 ± 0.92	0.02590 ± 0.001	<LOD
**S2 pH4 28 **	<LOD	19.66 ± 1.02	0.09770 ± 0.002	<LOD
**S2 pH4 42**	<LOD	19.98 ± 0.88	0.1044 ± 0.001	<LOD
**S2 pH7 1**	<LOD	20.05 ± 1.35	0.08210 ± 0.001	<LOD
**S2 pH7 28**	<LOD	20.45 ± 1.41	0.07540 ± 0.001	<LOD
**S2 pH7 42**	<LOD	20.35 ± 1.86	0.07230 ± 0.002	<LOD
* ** 10% Biomin F ** *				
**S3 pH4 1**	0.6543 ± 0.002	41.52 ± 2.02	0.6316 ± 0.03	1.849 ± 0.02
**S3 pH4 28**	0.8841 ± 0.003	49.70 ± 2.11	15.19 ± 0.92	0.138 ± 0.001
**S3 pH4 42**	1.3700 ± 0.001	52.44 ± 1.94	19.00 ± 0.76	<LOD
**S3 pH7 1**	0.6234 ± 0.001	42.01 ± 2.05	1.016 ± 0.09	2.665±
**S3 pH7 28**	0.5772 ± 0.002	47.16 ± 1.86	8.975 ± 0.08	0.3251±
**S3 pH7 42**	0.5800 ± 0.001	44.73 ± 1.76	7.771 ± 0.08	<LOD
* ** 10% Biomin C ** *				
**S4 pH4 1**	2.046 ± 0.03	39.56 ± 2.11	0.2752 ± 0.002	<LOD
**S4 pH4 28**	79.70 ± 2.02	34.02 ± 1.64	2.224 ± 0.09	<LOD
**S4 pH4 42**	62.75 ± 1.45	20.59 ± 1.29	3.414 ± 0.07	<LOD
**S4 pH7 1**	1.381 ± 0.03	40.92 ± 1.84	0.4609 ± 0.005	<LOD
**S4 pH7 28**	20.18 ± 0.18	10.18 ± 0.84	1.293 ± 0.07	<LOD
**S4 pH7 42**	20.14 ± 0.21	0.4463 ± 0.03	2.066 ± 0.05	<LOD
* ** 10% 45S5 ** *				
**S5 pH4 1**	1.216 ± 0.01	38.72 ± 1.22	1.466 ± 0.01	<LOD
**S5 pH4 28**	1.720 ± 0.02	40.19 ± 1.65	5.522 ± 0.01	<LOD
**S5 pH4 42**	2.750 ± 0.04	41.43 ± 2.23	5.763 ± 0.02	0.1745 ± 0.001
**S5 pH7 1**	0.9079 ± 0.01	39.80 ± 2.06	1.133 ± 0.05	0.1286 ± 0.002
**S5 pH7 28**	1.669 ± 0.03	38.71 ± 2.11	3.828 ± 0.03	0.1965 ± 0.001
**S5 pH7 42**	1.686 ± 0.03	38.78 ± 1.82	4.747 ± 0.02	0.8448 ± 0.002

LOD—below detection limit.

## Data Availability

The data supporting the reported results can be provided by the authors on request.

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
