# Peer review of "Mechanical Properties and Biocompatibility of 3D Printing Acrylic Material with Bioactive Components"

_jfb, 2022, doi:10.3390/jfb14010013_

Round 1
Reviewer 1 Report
(1) Please remove the “Type of the Paper ()” term.
(2) On page 2, line 50, ZrO2 should be revised. The same problem can be found with another chemical formula.
(3) The first paragraph of the introduction needs also the following reference: Additive manufacturing: an opportunity for the fabrication of near-net-shape NiTi implants, Journal of Manufacturing and Materials Processing 6.3 (2022): 65.
(4) The authors did not explain the novelty and significance of their work in the introduction part. Indeed, the introduction part is not cohesive. Topics change from sentence to sentence. The authors should follow the funnel procedure. The funnel technique for writing the introduction begins with generalities and gradually narrows your focus until you present your thesis.
(5) Please remove the MS word spelling checker redline from Fig. 1. A little line is also visible at the bottom of the image.
(6) Please provide proper references for equations on page 6.
(7) The purity and manufacturer of reagents should be given. Many essential details of surface characterization and evaluation are missing. The description is virtually reduced to listing the methods and instruments used, rather than details of procedures and subsequent analysis.
(8) Please use a scale bar for Fig. 3. Or, you can even use a ruler as a scale.
(9) Some formatting errors can be found at the end of page 10.
(10) The sentence “The additional release of cations can locally raise the pH above 5.5, which can greatly prevent the early stages of caries.” needs the following reference: International Journal of Molecular Sciences 23.7 (2022): 3665.
Author Response
Dear Editor,
We would like to respond to the comments of the Reviewers of the manuscript entitled ,,3D printing orthodontic material with bioactive properties“ Z. Raszewski, K. Chojnacka, J. Kulbacka and M. Mikulewicz, to the Journal of Functional Biomaterials. We included all the changes according to the suggestions of the Reviewers. We must admit that they were very useful and were a significant contribution to the manuscript. The Reviewers have read the paper thoroughly, and remarks were very helpful to improve the manuscript. Below, we would like to give answers to the specific points.
Reviewer 1
- Please remove the “Type of the Paper ()” term.
Thank you for the very good reviews, it has been corrected- type of paper was removed
- On page 2, line 50, ZrO2 should be revised. The same problem can be found with another chemical formula.
Thank you for the very good reviews, it has been corrected, in all chemical formulas
- The first paragraph of the introduction needs also the following reference: Additive manufacturing: an opportunity for the fabrication of near-net-shape NiTi implants, Journal of Manufacturing and Materials Processing 6.3 (2022): 65.
This very important article has been added to the introduction section
- The authors did not explain the novelty and significance of their work in the introduction part. Indeed, the introduction part is not cohesive. Topics change from sentence to sentence. The authors should follow the funnel procedure. The funnel technique for writing the introduction begins with generalities and gradually narrows your focus until you present your thesis.
Thank you for the very good reviews, it has been corrected. Part of the introduction has been changed:
“In recent years, the market has seen an increasing share of materials for the production of prosthetic restorations, which are printed by 3D printing. They can be used to make, among others, temporary crowns and bridges, surgical templates, removable orthodontic appliances, and dentures [1].
3D printing has undoubtedly many advantages over the traditional process of forming orthodontic appliances. First, there is no need to make models (additional material). In the thermoforming process, waste is additionally generated from the foil that has been pressing on the model. Furthermore, this new technology can shorten the time to make a single appliance and reduce laboratory labor costs [2].
However, a number of publications indicate that 3D printed materials have lower mechanical properties than products made with traditional heat curing resins or CAD CAM technology [3]. To solve this problem, investigations are underway in two main ways. The first direction is the development of materials for 3D printing consists of attempts to increase their mechanical properties by adding nanofillers, e.g., ZrO2 , [2,4] cellulose nanocrystals and silver nanoparticles [5,6].
The second activity focuses on the development and synthesis of new acrylic oligomers, which possess better mechanical properties and less shrinkage during polymerization [7].
In the literature, it is possible to find a number of information on the mechanical properties tested by the research of materials suitable for 3D printing and their biocompatibility. However, these are mostly commercial products, allowed for sale, but their exact composition remains the manufacturer's secret. Therefore, an important piece of information for the reader may be to perform a research model with methacrylate resins and test their properties [8].
Another important consideration for 3D materials is their contact time with the patient's oral cavity. Removable clear aligners are in direct contact with the patient's mouth for a long period of time (22 hours / day). If the patient does not clean his teeth after eating, ideal conditions are created for the demineralization of the enamel through the action of sugar-decomposing bacteria. For this group of users, it would be good if this material had bioactive properties. On the one hand, these properties prevent the formation of biofilms on their surface, but also provided the ions that could be used to initiate the remineralization process of the tooth that is in direct contact with the material [9]. Additional release of cations can locally raise the pH above 5.5, which can greatly prevent the early stages of caries [10].”
- Please remove the MS word spelling checker redline from Fig. 1. A little line is also visible at the bottom of the image.
It was done
- Please provide proper references for equations on page 6.
Thank you for the very good reviews, it has been added ref number [17]
- The purity and manufacturer of reagents should be given. Many essential details of surface characterization and evaluation are missing. The description is virtually reduced to listing the methods and instruments used, rather than details of procedures and subsequent analysis.
Thank you for the significant remark. The research methods used by us were presented in more detail in our previous publication. The purity of the raw materials has been entered in Table 1
- Please use a scale bar for Fig. 3. Or, you can even use a ruler as a scale.
Thank you for the significant remark. Figure 3 contains the scale bars which are in the upper left corner of each microphotograph corresponding to 500 µm
- Some formatting errors can be found at the end of page 10.
Thank you for the very good reviews, it has been corrected. It was our mistake when formatting the text
- The sentence “The additional release of cations can locally raise the pH above 5.5, which can greatly prevent the early stages of caries.” needs the following reference: International Journal of Molecular Sciences 23.7 (2022): 3665.
Thank you for the very good reviews, article has been added
Reviewer 2 Report
The title is too broad. Suggest for the authors to be more specific e.g. "Mechanical properties and biocompatibility of 3D printing acrylic material with bioactive components"
Abstract
The first sub-heading should be aims, not abstract. The study was assessing the properties of a resin material that incorporated bioactive components. It was misleading to direct it for use in orthodontics alone when in the introduction and conclusion it was suggested that such material can be used for other areas such as prosthodontics as well.
Line 99: please cite the previous publication.
Line 121: 10×3,3×64 . Please clarify the dimensions: Is this the area of two dimension or volume but a typo on 3,3?
Figure 1: Why does Biomin have a red underline?
Line 139:González, G et al. Please cite the reference number after this.
Line 141: diameter of 15 mm × 1 mm. I am not clear which is the diameter. 15mm or 1mm. Do you mean 1mm being the thickness?
Line 150: Articles describing 3D materials forward the thesis that the mechanical properties of the printed material depend on the orientation of the printed object: Please cite the articles to support this statement.
Line 164: Why was the dimension selected?
Line 184: Please state the brand and manufacturer of the desiccator. How many days did you wait to confirm constant?
Line 211: did you mean "chosen"?
Results
Table 2: I find this table difficult to interpret. What are the commas? What are the units for the values. How do you show differences between 10 mins and 30mins were significant?
Table 3: I find this table difficult to interpret too. I couldn't work out the p-values were showing differences between which cell.
Table 4: Similar problem with table 3.
Figure 3: I do not see the label a to c. So I can't work out which images they were referring to in the figure description.
Figure 3 and 4: suggest to place a scale.
Figure 5: Which were the tested material shown? There was no detail of this.
Figure 6 showed significant differences between the time points of the test products, but what was the post-hoc outcome? Given that changes over time were compared, shouldn't the authors do a repeated measures ANOVA rather than one-way ANOVA?
Table 5: I dont understand what 1, 8, 42 represent
Discussion
The bioactive materials were tested to determine the release for up to 42 hours only. What is the long term potential benefit when patients wear dental materials in their mouth over months or years without changing the appliances.
Conclusion
Should reflect the objective
Author Response
Dear Editor,
We would like to respond to the comments of the Reviewers of the manuscript entitled ,,3D printing orthodontic material with bioactive properties“ Z. Raszewski, K. Chojnacka, J. Kulbacka and M. Mikulewicz, to the Journal of Functional Biomaterials. We included all the changes according to the suggestions of the Reviewers. We must admit that they were very useful and were a significant contribution to the manuscript. The Reviewers have read the paper thoroughly, and remarks were very helpful to improve the manuscript. Below, we would like to give answers to the specific points
Reviewer 2
- The title is too broad. Suggest for the authors to be more specific e.g., "Mechanical properties and biocompatibility of 3D printing acrylic material with bioactive components"
Thank you for a very good title for this article, it has been changed according to your suggestion into:
“ Mechanical properties and biocompatibility of 3D printing acrylic material with bioactive components”
- Abstract
The first sub-heading should be aims, not abstract. The study was assessing the properties of a resin material that incorporated bioactive components. It was misleading to direct it for use in orthodontics alone when in the introduction and conclusion it was suggested that such material can be used for other areas such as prosthodontics as well.
Thank you for your attention. We wanted to list all possible applications of these types of materials. Both in orthodontic technique and prosthetic applications. The second thing is that the material as an orthodontic aligner is used by the shorts for a period of time and then it is replaced with a new one. For now, we have research showing that it can release ions for a period of 42 days. We would like to extend the research for a longer period of time, so that the material can be used differently in other areas of dentistry.
- Line 99: please cite the previous publication.
Thank you for the very good reviews, it has been fixed, publications number 14 and 15 were added
- Line 121: 10×3,3×64 . Please clarify the dimensions: Is this the area of two dimension or volume but a typo on 3,3?
Thank you for your comments, this is the length of the sample, width, and thickness, it has been corrected
- Figure 1: Why does Biomin have a red underline?
Samples stored in water were marked with a black felt-tip pen on the other side. Unfortunately, under the influence of hydrolysis, after the experiment it was not possible to remove the remnants of this marking. We show the hydrolysis after the tests to show how they changed under the influence of the test.
- Line 139: González, G et al. Please cite the reference number after this.
Thank you for the very good reviews, it has been corrected and publication number 16 was added
- Line 141: diameter of 15 mm × 1 mm. I am not clear which is the diameter. 15mm or 1mm. Do you mean 1mm being the thickness?
Thank you for the very good reviews, it has been corrected. Samples have diameter 15 mm and thickness 1 mm
- Line 150: Articles describing 3D materials forward the thesis that the mechanical properties of the printed material depend on the orientation of the printed object: Please cite the articles to support this statement.
Thank you for the very good reviews, it has been added publication number 3.
- Line 164: Why was the dimension selected?
Thank you for the very good reviews, it has been corrected. Dimensions of the sample according to the requirements of the ISO 20795-1: 2013 standard
- Line 184: Please state the brand and manufacturer of the desiccator. How many days did you wait to confirm constant?
Thank you for the very good reviews, it has been added, Samples were storage for 14 days. Producer of desiccator is Shott from England.
- Line 211: did you mean "chosen"?
It means -were selected for further research. Thank you for the very good reviews, it has been fixed
Results
- Table 2: I find this table difficult to interpret. What are the commas? What are the units for the values? How do you show differences between 10 mins and 30mins were significant?
It has been corrected. Columns indicate Absorption of individual components and standard deviation; statistical analysis was provided. Thank you for your valuable comment
- Table 3: I find this table difficult to interpret too. I couldn't work out the p-values were showing differences between which cells.
Thank you for the very good reviews, it has been corrected. It was our mistake when formatting the text.
- Table 4: Similar problem with table 3.
Thank you for the very good reviews, it has been corrected. It was our mistake when formatting the text
- Figure 3: I do not see the label a to c. So I can't work out which images they were referring to in the figure description.
Thank you for the very good reviews, it has been fixed. It was our mistake when formatting the text
- Figure 3 and 4: suggest placing a scale.
Thank you for the very good reviews, it has been fixed. The sample size is listed under Figure 3 and Figure 4.
- Figure 5: Which were the tested material shown? There was no detail of this.
Sorry, the caption under the picture was mixed during editing. It has been changed.. X axis from left - Background Resin, Resin +10% Biomin C. Y axis is 24, 48, 72 hours (from top left)
- Figure 6 showed significant differences between the time points of the test products, but what was the post-hoc outcome? Given that changes over time were compared, shouldn't the authors do a repeated measures ANOVA rather than one-way ANOVA?
Thank you for the very good reviews, it has been fixed. A new analysis has been performed.
- Table 5: I dont understand what 1, 8, 42 represent
Thank you for your attention, these numbers are days, 7,28,42 it has been corrected
Discussion
The bioactive materials were tested to determine the release for up to 42 hours only. What is the long-term potential benefit when patients wear dental materials in their mouth over months or years without changing the appliances?
Thank you for your important attention. Our work focused on the basic model, i.e. class 1 materials with a contact time of up to 30 days. Orthodontic aligners can be replaced after 10-14 days. In the next stage of our research, we want to extend the elution times.
Conclusion
Should reflect the objective
Thank you for the very good reviews, it has been fixed and objective of our research was added.
- The thesis presented at the beginning of our research has been confirmed. It is possible to make 3D printing material prepared with 10% addition of various active glasses, which can release calcium, silicon, and phosphate ions.
- In the case of Biomin F glass, the release of fluorine ions in an acidic and neutral environment was very dynamic (it occurred within the first 24 h).
- Acrylic resins modified with 10% Biomin C glass can be valuable sources of calcium cations and phosphate anions under acid and neutral conditions over a period of 42 days.
- More research is needed on the composition of the methacrylate resins used to create the material for 3D printing, which would have a higher mechanical resistance to fracture. This section is not mandatory but can be added to the manuscript if the discussion is unusually long or complex.
Reviewer 3 Report
Dear Authors,
the paper is really interesting, well conducted and fits the objectives of the journal; but it is necessary to review some points in order to improve the quality of the paper:
-First, i ask you to check the plagiarism of your article using specific sites to get a similitary report
-About the Title of the article,I suggest you to modify it and add the type of article.
The introduction section is very short and is needed to add other references to increase the quality of the manuscript.
Preferably a published articles should be with 90 or more references.
I suggest you some articles
Dento-Skeletal Class III Treatment with Mixed Anchored Palatal Expander: A Systematic Review DOI: 10.3390/app12094646
Perception about Health Applications (Apps) in Smartphones towards Telemedicine during COVID-19: A Cross Sectional Study. Doi: https://doi.org/10.3390/jpm12111920Teledentistry in the Management of Patients with Dental and Temporomandibular Disorders Doi: https://doi.org/10.1155/2022/7091153
Patient-reported outcomes while managing obstructive sleep apnea with oral appliances: a scoping review. Journal of Evidence-Based Dental Practice. 2022 Oct;101786. Doi: https://doi.org/10.1016/j.jebdp.2022.101786
-You need to review the grammar and English of your article
-I suggest you add a table with the list of abbreviations used in the text.
Regards
Author Response
Dear Editor,
We would like to respond to the comments of the Reviewers of the manuscript entitled ,,3D printing orthodontic material with bioactive properties“ Z. Raszewski, K. Chojnacka, J. Kulbacka and M. Mikulewicz, to the Journal of Functional Biomaterials. We included all the changes according to the suggestions of the Reviewers. We must admit that they were very useful and were a significant contribution to the manuscript. The Reviewers have read the paper thoroughly, and remarks were very helpful to improve the manuscript. Below, we would like to give answers to the specific points.
Reviewer 3
Comments and Suggestions for Authors
Dear Authors,
the paper is really interesting, well conducted and fits the objectives of the journal; but it is necessary to review some points in order to improve the quality of the paper:
1) First, i ask you to check the plagiarism of your article using specific sites to get a similitary report
Thank you for the very good reviews, it has been checked
- About the Title of the article, I suggest you modify it and add the type of article.
Thank you for the very good reviews, the title has been changed.
- The introduction section is very short and is needed to add other references to increase the quality of the manuscript.
Thank you for the very good reviews. New information and new literature items have been added to the introduction part.
In recent years, the market has seen an increasing share of materials for the production of prosthetic restorations, which are printed by 3D printing. They can be used to make, among others, temporary crowns and bridges, surgical templates, removable orthodontic appliances, and dentures [1].
3D printing has undoubtedly many advantages over the traditional process of forming orthodontic appliances. First, there is no need to make models (additional material). In the thermoforming process, waste is additionally generated from the foil that has been pressing on the model. Furthermore, this new technology can shorten the time to make a single appliance and reduce laboratory labor costs [2].
However, a number of publications indicate that 3D printed materials have lower mechanical properties than products made with traditional heat curing resins or CAD CAM technology [3]. To solve this problem, investigations are underway in two main ways. The first direction is the development of materials for 3D printing consists of attempts to increase their mechanical properties by adding nanofillers, e.g., ZrO2 , [2,4] cellulose nanocrystals and silver nanoparticles [5,6].
The second activity focuses on the development and synthesis of new acrylic oligomers, which possess better mechanical properties and less shrinkage during polymerization [7].
In the literature, it is possible to find a number of information on the mechanical properties tested by the research of materials suitable for 3D printing and their biocompatibility. However, these are mostly commercial products, allowed for sale, but their exact composition remains the manufacturer's secret. Therefore, an important piece of information for the reader may be to perform a research model with methacrylate resins and test their properties [8].
Another important consideration for 3D materials is their contact time with the patient's oral cavity. Removable clear aligners are in direct contact with the patient's mouth for a long period of time (22 hours / day). If the patient does not clean his teeth after eating, ideal conditions are created for the demineralization of the enamel through the action of sugar-decomposing bacteria. For this group of users, it would be good if this material had bioactive properties. On the one hand, these properties prevent the formation of biofilms on their surface, but also provided the ions that could be used to initiate the remineralization process of the tooth that is in direct contact with the material [9]. Additional release of cations can locally raise the pH above 5.5, which can greatly prevent the early stages of caries [10].
In materials that have a longer contact with the patient, a very important parameter is their biological compatibility (it must not be cytotoxic, irritating, or allergic) [11]. However, polymerization during 3D printing causes only partial reaction of the double bonds and a certain degree of conversion on the C=C double bonds in the methacrylic group. For this reason, for the material to be fully cured and more bio compatible, it is necessary to place the material in a light furnace for the post-curing process.
The curing time of the material after 3D printing is of great importance for its properties. It depends, among other things, on the color of the material, the single thickness of the printed layer, and the composition of the resin itself. Materials with acrylates in their composition undergo faster cross-linking reactions and show a lower value of oxygen inhibition [12]. Cycloaliphatic and aromatic acrylates show less shrinkage compared to the common monomer with low molecular mass, such as triethyl glycolmethacrylate or 2- hydroxy ethyl methacrylate. Shrinkage and associated stress can influence the deformation during 3D printing when the object is created layer by layer [13].
All these parameters must be taken into account when creating a new 3D printing material. During this research, it was hypothesized that there is a possibility of creating a material with the possibility of long-term ions releasing, which can be used in a 3D printer. Because in our previous work, we managed to create a material with to those of the materials similar properties that used poly (methyl methacrylate) as an organic matrix. As 3D printing materials contain higher methacrylates, it is necessary to confirm this hypothesis also for other compounds of this group [14,15].
- Preferably a published articles should be with 90 or more references.
I suggest you some articles
Dento-Skeletal Class III Treatment with Mixed Anchored Palatal Expander: A Systematic Review DOI: 10.3390/app12094646
Perception about Health Applications (Apps) in Smartphones towards Telemedicine during COVID-19: A Cross Sectional Study. Doi: https://doi.org/10.3390/jpm12111920Teledentistry in the Management of Patients with Dental and Temporomandibular Disorders Doi: https://doi.org/10.1155/2022/7091153
Patient-reported outcomes while managing obstructive sleep apnea with oral appliances: a scoping review. Journal of Evidence-Based Dental Practice. 2022 Oct;101786. Doi: https://doi.org/10.1016/j.jebdp.2022.101786
Thank you for the very good reviews, these important articles have been added
- You need to review the grammar and English of your article
Thank you for the very good reviews, it has been review.
- I suggest you add a table with the list of abbreviations used in the text.
Thank you for the very good reviews, the aberration list has been added to the end of the manuscript.
Thank you again for all your valuable comments. They have been added to the text
Abreviation
Aerosil R972- fumed silica commercial product from Evonik
CAD CAM- Computer-Aided Design/computer-Aided Manufacturing
DMEM- Dulbecco’s Modified Eagle Medium
E4GMA- Ethoxylated (E4) bisphenol A dimethacrylate
EDTA- Ethylenediaminetetraacetic acid
HGFs- human gingival fibroblasts
LOD- below the detection limit
PE- polyethylene
TEGDMA- Triethylene glycol dimethacrylate
TPO- diphenyl(2,4,6-trimethylbenzoyl) phosphine oxide
UDMA- 7,9,9-trimethyl-4,13-dioxo-3,14-dioxa-5,12-diazahexadecane-1,16-diyl bis methacrylate
Round 2
Reviewer 3 Report
Dear Authors,
i'm satisfied in my opionio th epaper is ready for the publications
regards